# Pulsing Addition to Modulated Electro-Hyperthermia

**DOI:** 10.3390/bioengineering11070725

**Published:** 2024-07-17

**Authors:** Andras Szasz

**Affiliations:** Department of Biotechnics, Hungarian University of Agriculture and Life Sciences, 2100 Gödöllő, Hungary; biotech@gek.szie.hu

**Keywords:** hyperthermia, tumor, electric-field, thermal, energy, pulse, nonthermal, apoptosis, cell-selection

## Abstract

Numerous preclinical results have been verified, and clinical results have validated the advantages of modulated electro-hyperthermia (mEHT). This method uses the nonthermal effects of the electric field in addition to thermal energy absorption. Modulation helps with precisely targeting and immunogenically destroying malignant cells, which could have a vaccination-like abscopal effect. A new additional modulation (high-power pulsing) further develops the abilities of the mEHT. My objective is to present the advantages of pulsed treatment and how it fits into the mEHT therapy. Pulsed treatment increases the efficacy of destroying the selected tumor cells; it is active deeper in the body, at least tripling the penetration of the energy delivery. Due to the constant pulse amplitude, the dosing of the absorbed energy is more controllable. The induced blood flow for reoxygenation and drug delivery is high enough but not as high as increasing the risk of the dissemination of malignant cells. The short pulses have reduced surface absorption, making the treatment safer, and the increased power in the pulses allows the reduction of the treatment time needed to provide the necessary dose.

## 1. Introduction

Hyperthermia as a cancer cure is one of the early medical practices that originated from ancient medicine. The medical processes using heat remain a vital “household remedy”, even nowadays. Electromagnetic heating techniques replaced the ineffective ancient heat delivery. The application of electromagnetic effects presented unique possibilities and renewed the hyperthermia methodology. In modern therapeutic practices, using electromagnetic processes (mainly radiation) to heat the whole body or its local volume developed rapidly over a century ago. Various technical solutions for oncologic hyperthermia (HT) attract growing attention among oncology professionals. 

The technical development of electromagnetic heating methods in the early 1900s revolutionized heat application for therapeutic gains, including malignancies. The curative processes with electromagnetic methods became available in the first quarter of the 19th century [1]. A French doctor, Arsene D’Arsonval, introduced a pure electromagnetic treatment called “Darsonvalization”. The absorbed electromagnetic energy resulted in heating. However, the physiological effects of heating (change in blood perfusion, thermal homeostatic regulations, risk of malignant dissemination, etc.) were initially neglected. The starting process was not free from extreme exaggerations. The German Electric Belt Agency went far, advertising that practitioners should reduce or even stop using drugs, advocating for electricity treatment alone [2].

The research on electromagnetic heating effects has revealed a complex interplay of factors. The temperature increase caused by electromagnetic energy absorption and the additional chemical changes (molecular excitations) induced by the electromagnetic field are key aspects. The thermal component is directly related to the square of the electric current, while the field component is proportional to that current. The bioelectromagnetic excitation alters the chemical bonds and the structure of compounds through direct electric forces, while some of the absorbed energy heats the target, raising its temperature. The thermal effect alone can also trigger chemical reactions, and the field excitation enhances these effects. This understanding has paved the way for therapeutic practices that combine electromagnetic molecular, cellular, and tissue excitation with heating.

A further push on the development was the discovery of microwaves, which began clinical practice working similarly to microwave ovens. The thermal effect of electromagnetic energy absorption was much more straightforward to understand and was more accessible to study; therefore, separating the heating and exciting (thermal and nonthermal) effects became dominant (Figure 1). Our present approach creates synergy between the thermal and nonthermal components of the electromagnetic energy absorption processes.

The method’s long history has not benefited it and has increased skeptical opposition, with varying positive and negative results. Infancy is standard for all developing systems but is abnormal when it is unusually long. Hyperthermia is mature for broad acceptance.

The currently known and accepted oncological hyperthermia effects can be divided into three categories, according to Figure 2:It destroys the tumor cells by absorbing energy;It has immunogenic effect;In the most frequent application, it sensitizes the conventional oncotherapies, like radiotherapy and chemotherapy.

Two primary electromagnetic therapies are applied in medicine: ionizing and nonionizing radiation. Ionizing radiation (e.g., radiotherapy) primarily has a nonthermal effect, breaking the DNA string with energy and causing more nonthermal radiative damage. At the same time, its thermal component is only tiny in most cases. The nonionizing radiation used in hyperthermia treatments is the opposite of the thermal/nonthermal ratio. It usually concentrates on the thermal component of the radiation, and the nonthermal component is a small part of conventional hyperthermia (Figure 3). The ratio of the thermal component effects looks at contrasts between the two methods, and direct cellular damage is also unlike. The direct cellular damage in the ionizing instance mostly breaks the DNA strands. At the same time, in nonionizing impact, the primary effect targets the cells as units, with concentration on the membrane damage at a temperature of about 42 °C. We discuss here the non-ablative applications, where the nonionizing has less energy than necessary for a direct burn.

## 2. Electrothermal Complexity

The electrothermal interactions are complex and can not be separated by thermal and nonthermal effects because their synergy does the job [3]. However, the ratio of these effects could be modified, forcing more nonthermal parts into the processes by the molecular excitation methods [4,5]. The primary selection mechanism uses the target’s extended electric and thermal heterogeneity, which can be utilized to select the molecular groups and excite the appropriate signal patterns [6]. The electric field gives the best opportunity for selection, which can technically be achieved with capacitive coupling [7]. Capacitive coupling with the intent of homogenous mass heating was applied on deep-seated [8] and superficial tumors [9]. Multiple clinical trials were performed for many metastatic stages. Soft-tissue [10] and Ewing sarcomas [11], pancreas cancer [12], breast cancer [13], liver tumors [14], rectal [15] and colorectal [16,17] malignancies, metastatic gastric tumors [18], urinary bladder lesions [19,20], esophageal [21,22,23], and head & neck tumors [24,25] all have shown the feasibility of capacitive hyperthermia with remarkable results. Some studies were performed on non-small-cell lung cancer, too [26,27], but its efficacy was questioned [28]. Intraluminal capacitive application for the esophagus has also shown feasibility [21,29].

However, numerous challenges need solutions. The physiological thermal homeostasis, the thermal and electric inhomogeneity of the tumor, the complexity of the bioelectromagnetic processes, which are induced by the hyperthermia, the controllable dose, and the nearly exponential decay of the energy absorption, which overheats the surface, are all complications that need to be solved. The conventional dose is connected to the tumor’s reached temperature, supporting isotherm homogeneity in the target. However, due to thermal and electric heterogeneity, the heating of biosystems is far from thermal equilibrium. Due to the physiological corrective feedback and the enormously inhomogeneous malignant target, fixing a homogeneous energy absorption in the tumor is impossible. The physiological feedback and thermodynamic processes destroy the possible homogeneous absorption part. The complex reality of living objects contradicts the macroscopic equilibrium. Moreover, the feedback processes deviate from the standard linearity of the specific absorption rate (SAR) and the temperature growth [30]. The isothermal expectation and the time-linearity in the heating process are only an illusion.

### 2.1. Temperature Development

The temperature development in the selected molecular group could be higher than their surroundings [31] and heat the area by the thermal convection of conduction, as is characteristically recognized in nanoparticle heating [32]. The extensively heated distance from the selectively heated small parts is small, less than 100 nm, and depends on how far it is from the artery, which cools it down. The heated molecular groups have relatively high temperatures due to the absorbed energy, and the larger volumes have gradually lower temperature averages (Figure 4). The concentrated energy absorption heats the transmembrane protein clusters (rafts), which heats the cell, but the average temperature will be less than in the raft. The cell heats the tumor, further lowering the average temperature. This thermal cascade makes it possible that the thermal effect, on average, remains safe and creates optimal conditions for the chemical reactions. Still, the micro-parts have enough energy to excite the necessary signal pathways.

The heating by the small protein molecular groups (rafts) has limitations. When the energy is too large, the transmembrane proteins are dehydrated and decomposed, which does not serve the signal excitation demand. The selection no longer works, and the thermal component starts to overdominate the entire process.

### 2.2. Selection

The first selection step is based on the metabolic differences between cancer and healthy cells. The reprogrammed metabolism (Warburg effect) increases the ionic concentration in the tumor microenvironment, so the well-chosen RF current may prefer to flow through the tumor. The next selective factor finds the tumor cells, which will differ from their healthy counterpart because of their autonomy and the broken healthy network, which provides differentiation in the dielectric constant of the cell and offers a cellular selection. The selected malignant cells have transmembrane protein molecular clusters (membrane rafts) that are the target of the selection. These rafts are embedded in the well-isolating lipid membrane, so their relative conductivity is high, and they massively absorb the energy [31]. The energy absorption with the well-chosen modulation, delivered by the RF carrier frequency, induces signals leading the cell to immunogenic cell death (Figure 5).

The impedance matching allows an energy-dose measure, ensuring that mainly the selected molecules absorb the energy. The specific absorption rate and temperature are connected at the physiological level by the blood flow (vasodilatation and vasocontraction changes) [30], which has no relevance when the heating is nanoscopic but has an increased modification of the energy intake by the homogenization of the temperature, leading to overdosing. Applied fractal modulation is also a novel technical innovation in impedance matching solutions [33], helping to produce immunogenic cell death (ICD) with its dynamic synchronization for homeostatic demands. The modulation with impedance-matching cellular selection completes the modulated electro-hyperthermia (mEHT) method [34,35]. The technical solution to optimize the ratio of the thermal and nonthermal components of the RF current needs a precise fit to the individual case, which changes with every treatment. The electronic solution is tuning the system to the measured impedance of the treated individual. The tuning seeks to form a touching situation as a purely metallic electrode would be fitted to the skin directly. This matching situation calculates the actual energy loss carefully, controls the reflected power, and matches the resonant compensation of the surface capacitor of the adipose tissue. The active impedance-guided capacitive solution (like mEHT) can use the bioelectromagnetic specialties of the malignant cells directly by RF current flow when it is matched to the optimized current. Cancer cells have an intensive metabolism to supply their proliferation [36]. The metabolic rate in most of the tumors is higher than their healthy counterpart (by at least 15% higher [37,38]), which selectively increases their temperature. The process has positive feedback because the growing temperature decreases the impedance of the tissue [39]. The high metabolic rate is used to identify the proliferation by positron-emission tomography (PET) [40,41]. The high nutrient/waste transport increases the ion concentration of the electrolytes in the surroundings of the malignant cell. The increased ionic concentration means a higher microenvironment conductivity [42] in the tumor cells and lowers the resistivity of the whole tumor. It could distinguish between healthy and malignant situations [43]. Due to the lack of a healthy cellular network in malignancy, the extracellular matrix of malignant cells has high dielectric permittivity, which can be used for selection [44,45]. The permittivity and the conduction modify the total impedance in the microenvironment of the malignant cells [46], which allows their automatic selection, while the RF current flows in the direction of the low electric impedance. The RF current density (specially chosen frequency and modulation) will self-selectively flow toward the malignant cells, which is measurable by the MRI current density image [47,48]. This effect is entirely automatic and follows any movements of the cells in real-time, solving the challenge of focusing.

The amount of apoptosis could be regulated by the ratio of the heating (temperature grows) and keeping (temperature is stable) periods. In the heating-up period, the thermal effect grows, and the excitation (non-thermal effect) grows due to the better conditions of the molecular reaction rates. In the stable temperature period, the thermal and non-thermal factors are constant, and the absorbed energy replaces the heat loss in the system. It is observed that during the heating-up period by mEHT, the apoptosis rate is significantly higher than that of the temperature-keeping period [49]. Applied step-up heating uses this difference to improve apoptotic processes [50].

### 2.3. Nonthermal Effects

Living objects are profoundly heterogenic. This heterogeneity defines the nonthermal electromagnetic interactions and the final effect. Electromagnetism acts through the various molecular and cellular structures, making energy absorption by current flow, making the polarization effect for polar molecules, exciting electrons between two energy levels, arranging the structure (order/disorder transition), making connected or separated clusters (percolation), breaking the cellular membrane (electroporation), inducing electrophoresis, electroosmosis, and excitation of the membrane channels. These effects are well-oriented, while the thermal effect primarily increases the kinetic energy of the molecules, which move faster and vibrate more intensely. The more considerable thermal energy may change the molecular interactions or cause phase changes (Figure 6).

The applied RF electric field changes the cells’ polarization and has different current components caused by the target’s impedance (Figure 7). The target’s capacitive behavior declines a part of the current from the ohmic component, which is mainly responsible for thermal effects by its vectorial direction.

Many molecular and physiological processes are determined by the heterogeneous lipid domains serving as molecular sorting platforms [51]. The malignant cells have a denser lipid-raft population on their membranes than their healthy counterparts [52]. Consequently, membrane heterogeneity is crucial in malignant cells’ selective energy absorption (Figure 8) and appears to be a synergy of thermal effects with nonthermal electricity.

### 2.4. Thermal Homeostasis

The thermal processes have a complex nonlinear interaction with homeostatic regulation, which tries to keep thermal homeostasis. The hypothalamus receives thermal signals (primarily from TRP receptors) and acts to reduce the temperature. The principal effects are blood flow with vasodilation and sweating trying to cool using an evaporating process. The thermal regulation balances the incoming energy, which nonlinearly fluctuates in the equilibrium (Figure 9) of incoming energy [59]. The fluctuation has various physiological components, including the opposition sensory mechanisms [60] and the various relaxation times of the different processes. The basic relaxation times could be measured with NMR [61], and the complex processes by various physiological measurements [62,63].

The temperature growth of local tissue in depth has a well-known character in time at constant absorbed power, defined by the specific absorption rate $\left(SAR=\frac{Absorbed\;power\;\left[W\right]}{Mass\;of\;absorber\;\left[kg\right]}\right)$. The absorbed energy ($E_{absorb}$) heats the local target, increasing its temperature (T). The physiological feedback process has a condition-dependent delay. Still, thermal homeostasis tries to restore equilibrium after the reaction time with intensive heat exchange and electrolyte transport ($E_{transport}$), like blood and lymph. Some absorbed energy also heats the surroundings, initially nontargeted issues ($E_{tissue}$). Consequently, the energy which increases the temperature ($E_{T}$) is less than the incident $E_{absorb}$ value. The absorbed energy grows the target temperature energy, but a part of the energy is used up for thermal homeostatic control (primarily the blood flow regulation) and heat conductivity to the neighboring tissues:

Considering the energy balance, the Pennes equation [64] describes the heating process: (1)$$\rho_{h}c\frac{\partial T}{\partial t}=\rho_{h}{SAR-c}_{b}\rho_{b}w_{b}\left(T\right)\left(∆T\right)-k_{h}\nabla^{2}T+q_{0}\rho 1.1^{\left(∆T\right)}$$

The terms in the Equation (1) are as follows: $\rho_{h}c\frac{\partial T}{\partial t}=\begin{array}{c} Energy\;for\;the\\ temperature\;grows\\ in\;the\;target \end{array}$, $\rho_{h}SAR=\begin{array}{c} Absorbed\;energy\\ from\;outside\\ incident\;power \end{array}$, $c_{b}\rho_{b}w_{b}\left(T\right)\left(∆T\right)=\begin{array}{c} Energy\;loss\;by\\ homeostatic\;regulation\\ \left(blood,\;lymph\;transport\right) \end{array}$, and $\left(k_{h}\nabla^{2}T+q_{0}\rho 1.1^{\left(∆T\right)}\right)=\begin{array}{c} Energy\;loss\\ by\;tissue\\ environment \end{array}$.

The c, ρ, and w are the specific heat, density, and perfusion. The subscripts denote the healthy tissue (h) and the blood (b) values. When the Pennes equation is applied to the tumor, the subscript will be t. The $w_{b}$ is the blood perfusion rate and T and t are the temperature and time, respectively. The analytical solution of this partial differential equation is a difficult task. The first approach uses the Green function [65,66] and the Green heat kernel function [67] and uses an analytical solution [68]. The point source Green function solution [69] can simulate the nanoparticle or thin needle with very local heating.

The differential equation can be well approached with differences when the time changes. The homeostatic control is slow enough and does not allow rapid changes, so the differences in the values can be used to approach the solution of the differential equations with enough accuracy, neglecting the minority effects in (1) at constant SAR. Introducing the temperature difference instead of the differential, $\Delta T=T-T_{b}$, we obtain the following:(2)$$\frac{\partial \Delta T}{\partial t}+\frac{1}{\tau }\Delta T=\frac{SAR}{c}\;$$
where $\tau_{0}≅\frac{1}{c_{b}w_{b}}$ is the relaxation time in the perfusion model according to the Pennes Equation (1). The solution to this difference-equation is as follows:(3)$$\Delta T=\frac{SAR}{c_{b}w_{b}}(1-e^{-\frac{t}{\tau }})$$

We must approximate the relaxation time in cancerous tissue. In rough approximations, the blood perfusion of the tumor is 0.833 $kg/sm^{3}$ when the temperature is below 41 °C and 0.416 $kg/sm^{3}$ when the temperature is above it [70]. Therefore, the following applies:(4)$$\begin{array}{c} \tau_{<41}≅\frac{10^{3}}{0.833}=1200s=20\;min,\\ \tau_{>41}≅\frac{10^{3}}{0.416}=40\;min \end{array}$$

However, the inflammatory reaction occurs in the surrounding tissues and not in the tumor itself, so the relaxation time in this constant perfusion model is as follows:(5)$$\tau_{0}≅\frac{10^{3}}{4}≅4.2\;min$$

It is the “wash-out” time of the heat perturbation in the tissue [71], depending on the blood flow of the studied tissue. This is the tissue relaxation after a heat shock by the blood-flow washout process. Healthy tissue is measured at ~4−7 min. The clinical standard average of SAR in the MHz range is 6 min [72], and we use it as a standard physiological average relaxation time, $\tau_{ave}=6\;min$.

The time of thermal washout could be modified by changing the metabolic rate by lowering the temperature, and it will make the tail of the washout function longer in time. Consequently, a longer $t_{0}^{\prime }>t_{0}$ a value will be added to the simple exponential, which depends on the decreased metabolism by the cooling process. This additional effect will cause a time lag because of the actual physiological time of the metabolic reaction. Due to the physiological self-time, which is about the same as the thermal washout physiological time, is approximately 6 min.

Thermal homeostasis regulates the temperature of the tissue. The absorbed energy drives temperature growth, but the increased blood flow, as an energy sink, counteracts. Over time, thermal homeostasis fixes the temperature in the thermal equilibrium, forming a steady-state process. The temperature, in this case, becomes constant. The absorbed energy substitutes the losses by $E_{transport}\;and\;E_{tissue}$, without changing the temperature (Figure 10).

All these dynamic changes depend sharply on personal homeostatic regulations, the electrolyte transport in the targeted volume, and the incident power. The dynamism of the heat delivery may also profoundly change the heating process [69]. Due to the physiological changes (first due to the blood-stream variation), the linear dependence breaks (Figure 11). When the SAR value is moderate, thermal surveillance develops an equilibrium [73]. However, a higher SAR (rapid temperature gain) could cause overshooting, compensated only after an overshooting, but the SAR could be as much as the homeostatic regulation cannot fix the temperature. Notably, the tumors are usually hotter than their host due to the high proliferation and energy use, even in thermal equilibrium.

The blood delivers the drugs for chemotherapies and oxygen for radiotherapy, so its behaviors are essential. The timing of the deliveries and washing out the toxic species are important factors to consider. In hyperthermia applications, the thermal washout time is driven by the BF and differs from the nonthermal clearance of molecules (like radiofarmacons, tracers, blood-delivered molecules, drugs, particles, or cells) from the tissues. The main difference is in the mechanisms of diffusion, which are different for various blood-delivered particles or molecules and heat. The thermal washout is also a complex process mainly driven by the BF, but not determined by it alone. The investigations of the clearance of tracers clearly show that the clearance (wash-out) tightly depends on the BF but these parameters are not equal; instantaneous mixing with metabolic changes and diffusion breaks the unity. Also, the metabolic heat does not directly affect the clearance, while the thermal washout is directly modified by it.

A “similarity” could be observed in the washout of tracers [74], which is a rescaling of the time, showing a similar scaling behavior as we saw in the heat-up process. The washout scaling “similarity” is also present in the wash-in of the tracer [75]. An important observation in contrast material studies is that the enhancement of the contrast material decreases with the temperature growth while increasing with the thermal cooling coefficient [76]. The main message is the high variability of the BF with tumor entities, and the tumors have a massively heterogeneous BF, having a gradient from the center to the periphery.

Hyperthermia protocols usually apply step-up heating specialized to the patient’s sensing. The heat pain effectively limits the hyperthermia dose. The patient senses the process and thus guides the personalized homeostatic heating-up dosing. It is more patient-friendly, causing as little discomfort as possible because the patient’s homeostatic control is active. The central task is to provide the proper dose. The actual protocol for the treated patient must be optimized to the given conditions and curatively effective with a high standard for safety, limiting the applied dose. This concept is entirely different from the conventional hyperthermia goals because instead of trying to produce isothermal volumes (equal temperature in the tumor), it uses heterogenic heating, following the heterogeneity of the tissue itself. This far-from-equilibrium heating keeps the driving force between the heated membrane rafts and its environment, pumping the heat from these nanoclusters to the cell interior.

When the intended dose is too much, it has to be corrected via personal notes. On the other hand, when the protocol presets a low energy dose, higher energy can be applied until the patient indicates the personalized limit. Overheating is practically impossible because the skin’s surface has the highest thermal load, and heat sensing is also there. This personalized dose regulation is the main factor for safety and, together with this, for efficacy. In proper step-up heating, no continuous increase of the temperature is applied. The primary governing process is homeostasis, so the heating fits that equilibrium. A steady-state gradual heating is necessary. The physiological response time must be considered. This characteristic time is when the homeostatic equilibrium is re-established in the new conditions after a definite disturbance. The average wash-out time in humans is approx. five to seven minutes. Considering the transient “break” of six min, the step-up heating is shown in Figure 12. A detailed calculation shows the rise in temperature and its dependence on the power function in step-up heating [49,50].

## 3. Semi-Adiabatic Synergy (SAR)

At the start of the heating, when the physiology does not impact ($E_{transport}=E_{tissue}=0$), the complete $E_{T}$ makes a temperature increase, so in this case, the following applies:(6)$$E_{T}=E_{absorb}=m_{t}\cdot c_{t}\cdot ∆T$$
where $m_{t}$ is the heated tumor mass, $c_{t}$ is the specific heat of the tumor, and ∆T is the temperature increase. In a shorter time than the actual relaxation of the tissue, the thermal homeostasis does not modify the absorbed energy, and the SAR directly defines the temperature development. Using (6) in this initial stage as follows:(7)$$SAR=\frac{E_{absorb}}{m_{t}\cdot ∆t}=\frac{m_{t}\cdot c_{t}\cdot ∆T}{m_{t}\cdot ∆t}=c_{t}\cdot \left(\frac{∆T}{∆t}\right)$$

This initial period is too short, causing a physiological reaction with blood perfusion. It is “semi-adiabatic”, and only the SAR acts for temperature development, neglecting the thermal homeostatic processes led by blood perfusion, but additionally keeps away from the growing metabolic rate, the change of the absorbed power due to the variation of electric and thermal parameters with the temperature, etc. All the absorbed energy appears like it would be a non-living target, so the thermal and nonthermal synergy appears. Due to this period not dealing with the homeostatic processes, the nonthermal effects occur more than in the further heating when the thermal control works. So, the nonthermal impact is dominantly active in this period Figure 13.

The temperature profile after this semi-adiabatic period declines with the slope of the temperature change and seeks thermal equilibrium without changing the temperature (Figure 14). This equilibrium temperature requests an energy dose, which replaces the lost energy by cooling.

The equilibrium process could be described with a stochastic explanation [49], approximating the different heat transfer parts. When thermal homeostasis stabilizes the temperature, the absorbed energy replaces that lost by heat conduction, convection, and radiation to keep the equilibrium. When the temperature development deviates from the slope, going stationary, another so-called constant perfusion rate model could be introduced. We seek to equilibrate the dominant factors that remain in the Pennes Equation (1) as follows:(8)$$\rho_{h}c_{h}\frac{\partial T}{\partial t}=\rho_{h}{SAR-c}_{b}\rho_{b}w_{b}\left(T\right)\left(∆T\right)$$

The solution of (8) is as follows:(9)$$∆T=\frac{\rho_{h}SAR}{{c_{b}\rho }_{b}w_{b}\left(T\right)}\left(1-e^{-\left(\frac{t}{\tau_{0}}\right)}\right)$$
where
(10)$$\tau_{0}=\frac{{c_{h}\rho }_{h}}{{c_{b}\rho }_{b}w_{b}\left(T\right)}≅\frac{1}{w_{b}\left(T\right)}$$
is the time constant of the constant perfusion model. Using realistic parameters, we obtain $\tau_{cp}≅6\;min$. When t is large, the system reaches thermal equilibrium, and no further rise in the temperature is observed:(11)$$0=\rho_{h}{SAR-c}_{b}\rho_{b}w_{b}\left(T\right)\left(∆T\right)$$

From (11), the equilibrium temperature is as follows:(12)$$T_{eq}≅1.4\frac{SAR}{{c_{b}\cdot w}_{b}\left(T_{eq}\right)}=1.4\frac{SAR}{perfusion}$$

When the power is switched off, the target cools down, determined by the wash-out time.

The impedance matching covers the cooling process and stabilizes the homeostatic control in the subcutis layer under the electrodes. Significant differences appear in the doses during the heating period, showing an increase in the temperature until it reaches the stable thermal equilibrium. The emphasis on the thermal or nonthermal processes makes the principal difference between the two heating periods. The nonthermal period primarily depends on the heating technique, the position of the tumor, and the initial power density $\left[\frac{W}{{cm}^{2}}\right]$. The semi-adiabatic period in radiation is between 10 and 15 min [77], which is about 20% of the complete session time. However, some protocol modifications could change the ratio. Step-up heating typically increases relative nonthermal dominance because the power proceeds before the thermal regulation becomes active. Of course, the thermal and nonthermal effects are strongly synergistic and tightly interact (Figure 15).

The length of the semi-adiabatic synergy (SAS) period depends on many factors. In addition to the leading thermal homeostasis, the heterogeneity of the target and the dynamism of the forced energy absorption define the length of the semi-adiabatic period (Figure 16). When the heating has no loss by various heat exchanges, the slope of the temperature change is linear. In heterogeneous heating, the small parts of the material absorb most of the energy, gradually heating the volume. In this case, a shorter period remains adiabatic, and the heat conduction actively spreads the energy. When the system has no other energy losses (well isolated), the temperature growth has no equilibrium. When the thermal homeostatic dynamism is active, the heat is gone quickly and forms an equilibrium.

### 3.1. Semi-Adiabatic Synergy (SAS) Promotes Apoptosis

Two main categories (and many of their variants) cause cell death. Necrosis is the sudden rupture of the cell. The cytoplasm and the cellular organelles located freely in the extracellular matrix could cause inflammation or, in large quantities, could be toxic. Another major variant is apoptosis, where cell death is gentler. The cellular components became fragmented and could be embedded in lipid membranes for safe elimination. An extremely gentle fragmentation happens in immunogenic cell death when damage associated with a molecular pattern is released in an undestroyed form [78]. The unhurt molecules deliver information about the genetic structure of the cancer cell, which could be used to adapt the available dendritic cells for immune surveillance to overcome the evading capability of cancer [79,80]. The tumor-specific adaptive immune T cells (killer and helper cells) perform immune attacks on distant metastases to where the bloodstream delivers them [81], which was observed in human studies as well [82,83].

Nonthermal activity was widely studied [84,85], and its synergy with a thermal component of absorption is also studied [4,5]. The synergy of the thermal and nonthermal effects recognized in mEHT increases apoptosis compared to the only thermal conditions [86,87,88], which has been shown to be complementary to chemotherapy [89] and radiotherapy [90]. The effect of modulation as a purely nonthermal impact also increases apoptosis [91].

### 3.2. In Vitro Verification

In cell-line experimental conditions [49], it was proven that the increased temperature is proportional to the absorbed power (SAR) in semi-adiabatic states as shown in (7), while exponential declines from this linearity, as in (9), and equilibriums appear at constant temperature (12). The SAS conditions have nonthermal dominance. Apoptosis during this linear changing period may be compared to the equilibrium to study the thermal and nonthermal impact difference on the cellular level [49]. The experiment measured apoptosis during the two definite periods of energy absorption. The regular mEHT of the adenocarcinoma human alveolar basal epithelial cell line, A549 treatment, was performed at 42 °C starting from 25 °C. The heating details were studied by divided periods, as shown in Figure 17:Phase 1. heated the cells from the room temperature (25 °C) to the usual starting temperature of the in vitro experiments, the human body temperature (37 °C);Phase 2. heated the cell culture from 37 °C to 42 °C, which is the equilibrium temperature of many standard hyperthermia treatments expecting the thermal impact;Phase 3. kept the equilibrium 42 °C for 30 min;Phase 4. continued the equilibrium heating at 42 °C for the next 30 min.

The SAS period was 15 min, (with 18 W power) the equilibrium period 30 min (7.5 W). In this case, the apoptosis (measured by Annexin V positive cells) was 31.18%. Remarkable apoptosis was measured in phases I and II, while in the longer phases III and IV, the apoptotic activity was low. Replacing the 1st phase with purely thermal water bath (WB) heating, apoptosis decreased to 22.6%, and when the 2nd phase was also replaced with WB, apoptosis decreased further to 7.2%, showing that the non-thermal effect in the heating up SAS period had produced the majority of apoptosis (Figure 18). Counting that the SAS had 18 W for 15 min (16.2 kJ), equilibrium needs only 7.5 W of power for 30 min (13.5 kJ). The energy-corrected expected apoptosis in the equilibrium period increased a little and became ~8.64%, but much smaller than the apoptosis in the SAS period. With the WB (42 °C) and the incubation (37 °C), applied for the same time when the mEHT treatment was performed, apoptosis was low at 2.61% and 2.42%, respectively.

Numerous control experiments were performed [49], substituting the various phases with WB or incubation; the results showed the same dominance of the SAS period in apoptotic production. When the treatment time in the equilibrium temperature (42 °C) was doubled (added Phase 4.), apoptosis grew significantly from 31.18% to 31.63%. However, when the equilibrium period cooled down the cell line to 37 °C for 5 min and up again to 42 °C in the next 5 min, apoptosis significantly increased to 51% altogether (Figure 19). In this experiment, the pulsing increased apoptosis to more than two times more than the standard mEHT at the same time and same temperature. This remarkable result gave rise to the idea of the pulsed mEHT development.

The pulsing role in apoptosis well supports the importance of the semi-adiabatic heating period. At the same time, it proves the decisional role of the nonthermal effects on apoptosis. These experiences were verified in vivo.

### 3.3. In Vivo Verification

The in vivo rat model used immunocompetent animals [92]. An RG2 [D74] (ATCC^®^, CRL 2433™, Manassas, VA, USA) astrocytoma [93] was inoculated into the parietal lobe of syngeneic Fischer 344 rats. The inoculation was syngeneic, genetically sufficiently identical, and immunologically compatible to allow for transplantation. There were three groups (three animals in each): (1) sham, (2) continuous mEHT, and (3) periodically stopped, pulsed mEHT treatments. The pulsing periods were 6 min with a 0.5 duty cycle, using the homeostatic relaxation time shown in Figure 20. A gadolinium-based MRI contrast agent (MAGNEVIST^®^, 0.5 mmol/mL, 0.2 mL/kg bdw) was used to detect lesions associated with an altered blood-brain barrier, and the volume of the tumor was quantified at the 8th and 15th days after inoculations. The brain temperature was evaluated indirectly by measuring the temperature in the middle ear and using a correlation curve set up in an earlier experiment.

The tumor growth rate between the 8th and 15th days after inoculations was, in the case of sham animals, 23.73 ± 12.15, in the treated with classical mEHT protocol, 19.08 ± 0.49, and in the treated with pulsing mEHT protocol, 6.83 ± 2.02 (Figure 21).

The immunohistochemical analysis shows the highest effects of the extracellular release of HSP70 in pulsed treatment. The extracellular HSP70 molecule has a decisional role [94] in tumor-specific immune reactions, delivering information for antigen-presenting and killer T cell priming [95,96]. The reduction of the Ki67 protein, which marks proliferation, shows the suppression of the malignant activity in the rat highest with pulsed mEHT, Figure 22.

## 4. Pulsing Modulated Electro Hyperthermia

It is early knowledge that the blood flow and the speed of the heat delivery are connected. Rapid heating well differentiates the blood flow between the tumor and healthy host, while at slow heating, this difference tends to disappear [78], even when both were performed for 20 min at 43 °C. In an early experiment, a heating pulse (45 °C, 10 min) was used to treat experimental rhabdomyosarcoma BAll12 cells before continuous hyperthermia exposure for 3 h at 42.5 °C [97]. The starting pulse had a noticeable role in the results of the hyperthermia procedure. These observations emphasize the role of the semi-adiabatic heating period as one of the factors in the selection. Pulsed hyperthermia in cancer treatment refers to a technique where heat is applied to tumor tissue in short, controlled bursts rather than continuously. This approach can potentially enhance the effectiveness of cancer treatments while minimizing damage to healthy surrounding tissues. Pulsing is not a new heating technique. The precise temperature adjustment often uses pulsing for preciosity, like in the incubators [98]. Hyperthermia in cancer therapy also uses thermal cycling to enhance even the anticancer effect of natural compounds in pancreatic malignant cells [99], and it was also used in human treatment [100].

Continuous hyperthermia can also damage healthy tissues surrounding the tumor. Pulsed heating addresses this concern by delivering heat in cycles, having a heating phase (applied for a short duration and raising the temperature to the desired level), and following it with the resting phase (heating is stopped, allowing the tissue to cool down partially). This cycling provides for the following:Reduced risk of damage to healthy tissues: since healthy tissues cool down faster than tumors, they experience less heating during the resting phase;Potentially enhanced tumor damage: some tumor cells might be more susceptible to heat when exposed to pulsed heating than continuous heat;Improved treatment tolerability: patients may experience fewer side effects due to reduced overall heat exposure.

The nonthermal effects of pulsed electromagnetic fields have also been investigated [101] and are shown to have a harmful nonthermal effect when the pulsing power is high. Damage by the pulsing heat treatment is also displayed in skin layer experiments [102] and well applicable for the chemical and thermal activation of the TRPV3 vanillin receptor [103], which otherwise has a role in immunogenic effects in mEHT applications because the mechanism of these channels could be modified by electric field [5]. An essential application of the pulsed electric treatment makes the transient reversible opening of the blood-brain barrier (BBB) [104,105,106]. This non-invasive method could be applied to promote drug delivery to the brain, which the BBB blocks. The opening mechanism is connected to the impact on tight junctions even without thermal effect [107]. When the power has a periodic increase, the applied step-up heating is also a pulsed process with very low frequency. The pulsed electric field (PEF) heating has particular applications in pain management [108,109]. The oncological applications have just started and have a good perspective [110]. The pulses are short (most nanoseconds) [111]. Longer heating times (50 s) by magnetic pulses shows the usual heating pattern for breast cancer hyperthermia treatment [112] and for induced damage of the tumor microcirculation. Electrochemotherapy (ECT) is developing on this basis. Continuous ECT (galvanic treatment) is an old therapy [113], also developed in the early 1990s in Bad Aibling (Germany) [114]. The PEF application is quickly developing in electroporation [115]. While the “pulsing technique” is commonly used in hyperthermia treatments, capacitive hyperthermia specifically does not utilize this technique broadly.

The new mEHT with a pulsing technique is thermally assisted and provides all the advantages of continuous mEHT operation. The above-described selective mechanisms target the membrane rafts and the membrane microdomains, which have a role in intracellular signal excitation and regulation, as well as the ICD processes. The targeting of membrane rafts helps to develop novel complementary therapies to increase the sensitivity to chemotherapeutic compounds, opening the gate for drug penetration into the cell and reducing multidrug resistance [116]. The capacitive coupling could also give a unique advantage in modifying the membrane voltage [117] with a low electric field.

The personal sensing homeostatic step-up heating solves safety problems when the patient communicates about identifying any discomfort. The question naturally arises about the reliability of subjective sensing, but personal sensing is the best available method for monitoring the heating process. Personal sensing is typically used to drive many protocols active in today’s medical treatments. When the patient cannot tolerate the prescribed dose, it is lowered, trying to fit it to the personal tolerance level. There is no reliable personalized dosing without controlling the guidance of personal sensing. In pulsed heat treatment, immense power can be tolerated for the short pulsing time when the time between the two pulses is long enough to return the temperature to the tolerable zone, at least partially. In these conditions, personal sensing will be the average time of the power. The slow thermal and physiologic reaction to the rapid power absorption makes averaging possible. By applying heat in pulses, healthy tissue surrounding the targeted area has more time to cool between pulses, potentially minimizing damage and side effects. This can be especially beneficial for tumors located near sensitive organs or nerves.

### 4.1. The Pulsing Technique

The duty cycle is as follows:(13)$$D=\frac{\langle P\rangle }{P_{p}}=\frac{t_{p}}{t_{r}}$$
where $\langle P\rangle$ is the average, $P_{p}$ is the pulsed power, $t_{p}$ is the pulse width, and $t_{r}$ is the repetition time. While the pulse “window is rectangular, the resulting heating pattern differs. It all depends on the thermal parameters of the targeted mass (Figure 23).

When the duty cycle is low, the average temperature is unchanged and remains on the baseline, having enough time to cool between the pulses. However, when the duty cycle grows, the cooling down period relatively shortens and could not be enough to reach the baseline again. In this case, the average temperature rises. The growing temperature follows roughly a cumulative Weibull function [49] $W=exp\left(-{\left(\frac{t}{t_{0}}\right)}^{n}\right)$, where n is the shape parameter, and $t_{0}$ is the time parameter. Both depend on the target material (Figure 24).

### 4.2. Advantages

The pulsing of mEHT has numerous advantages, as follows:Enhanced Efficacy:
The active factor of the mEHT is the RF current, which selectively flows through the target. In pulsing conditions, the extreme power gives a proportionally sizeable current density, which causes an effect. Pulsed heating can be more effective at killing cancer cells than continuous heating at the same average temperature;The pulsing induces the semi-adiabatic start of temperature growth, which accounts for the most significant part of apoptosis and induces immunogenic processes;In a low-duty cycle, increased blood flow to the tumor during the off pulses can bring in more oxygen and nutrients needed for the heat to damage the cells.The high-power pulse may induce reversible electroporation, increasing synergic efficacy with complementary chemo and immune therapies;Pulsed heating does not drastically influence homeostatic regulation as continuous heating does. So, the treatment and natural regulations are more effective in cooperative harmony;Studies show that pulsed electric fields effectively relieve pain, improving patients’ quality of life.Control and Flexibility:
Pulsed heating allows for finer control over the temperature delivered to the target area. The pulse duration, frequency, and power can be adjusted to achieve the desired therapeutic effect while minimizing the heating of surrounding tissue;The power in the pulses may be kept constant; only the duty cycle changes the average power, which determines the temperature. Like digital technologies, the continuous power (and constant energy absorption in a pulse) makes the dose more controllable;The associated side effects are reduced due to the pulsed heating and its relatively long relaxing time with a low-duty cycle;The synergy of the thermal and nonthermal electric absorption is more reliable;Despite the large pulse power, skin and adipose burns are less likely because the subcutaneous blood flow is active, may quickly reduce the heat stress in the pausing period, the pulse is short to burn, and the low-duty cycle ensures the low average temperature on the surface, too.Potential Dose Reduction:
Due to potentially higher efficacy, pulsed heating might require lower overall heat doses than continuous heating, potentially reducing treatment time;The specific benefits of pulsed heating may vary depending on the type of cancer, tumor size, location, and other factors.Technical advantages:
The reduced cooling facility makes designing a simpler and more efficient electrode system possible;Forcing step-up heating is unnecessary; choosing the semi-adiabatic phase is automatic and self-adjusted;The tuning is more accessible because the power (the pulse intensity) is constant during all the processes;Having 200 W in continuous heating, the power is at a 36 cm depth (the heaviest patient) and is ~15% of the incident field, which is ~30 W. In the pulsed case, reaching the same temperature with 800 cap W pulses with 0.24 duty cycle (average power is also 200 cap W like it was in the continuous case), the power at 36 cm will be 120 W in pulses, which is a significant increase. I propose the idea that this method treats all depths in humans equally.

### 4.3. Limitations

While pulsed mEHT shows significant advantages in cancer treatment, there are still potential limitations and adverse effects. The severity of the possible adverse effects depends on various factors, such as individual health, treatment parameters, and tumor characteristics.

Local tissue damage: Although pulsed heating reduces overall heating time, localized areas within the treatment zone may still experience high temperatures, potentially leading to extended tissue damage. The selection mechanisms on the tumor localize it, so the host tissues are likely safe. Still, we may lose part of the immunogenic advantages by the necrotic way of tumor cell death;Pain: The heating process can cause discomfort, and individual sensitivity varies. Some patients might experience more intense pain with pulsed heating than continuous heating, but the overall pain reduction after the treatment likely works for all;Nerve effect: Depending on the location, pulsed mEHT could lead to local numbness, tingling, or other nerve-related issues, depending on the patient’s state;Systemic effects: Like any hyperthermia treatment, pulsed heating can cause systemic effects like thirstiness, fever, chills, fatigue, nausea, and vomiting. The severity of these effects depends on factors like individual health, treatment parameters, underlying medical conditions, and complementary medical applications;Tumor-specific risks: The significantly high-power intensity in the pulses may cause rapid tumor lysis syndrome, which is toxic;Unforeseen complications: As with any new medical technology, unforeseen complications are always possible. More research is needed to fully understand the long-term effects and potential rare side effects of pulsed mEHT. Open communication and regular monitoring during treatment are crucial to identify and promptly manage any adverse effects;Technical challenge:
The pulsing power and temperature averages could differ depending on the tumor’s thermal washout physiology, which patients may have differently.The average power depends on the duty-cycle, so it does not serve as a dose in the mEHT as it was in continuous power. The dose could be only the integrative absorbed energy.The pulsing can change the original 1/f modulation depending on its duty cycle.

## 5. Conclusions

The pulsed mEHT treatment gives additional advantages to the standard modulated electro-hyperthermia. This type of treatment provides promising improvements in terms of safety and efficacy. The additional impulse modulation (high-power pulsing) increases the mEHT efficacy. The increased penetration depth supports the treatment of deep-seated tumors for heavy patients, and the decreased thermal load on the skin and the adipose tissues increases the patient’s safety and quality of life

## Figures and Tables

**Figure 1 bioengineering-11-00725-f001:**
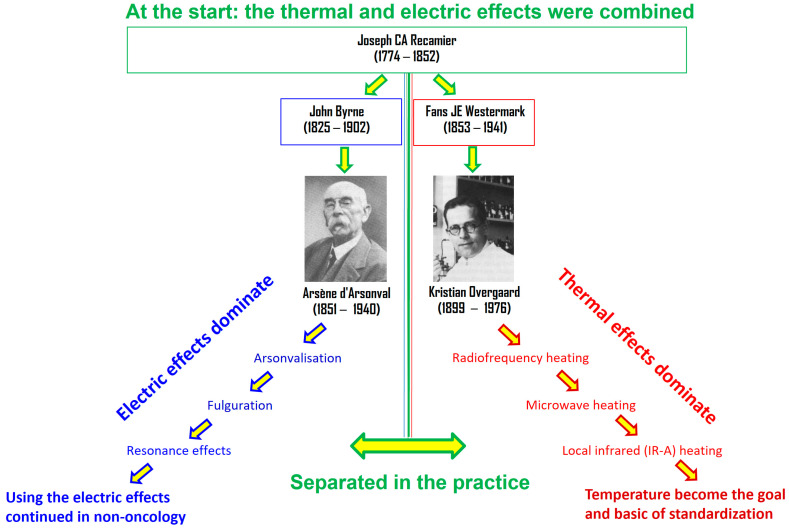
Arsene d’Arsonval (French) and Kristian Overgaard (Danish) are the leading doctors who divide the nonthermal field effects from heat, and the separation has widened over time.

**Figure 2 bioengineering-11-00725-f002:**
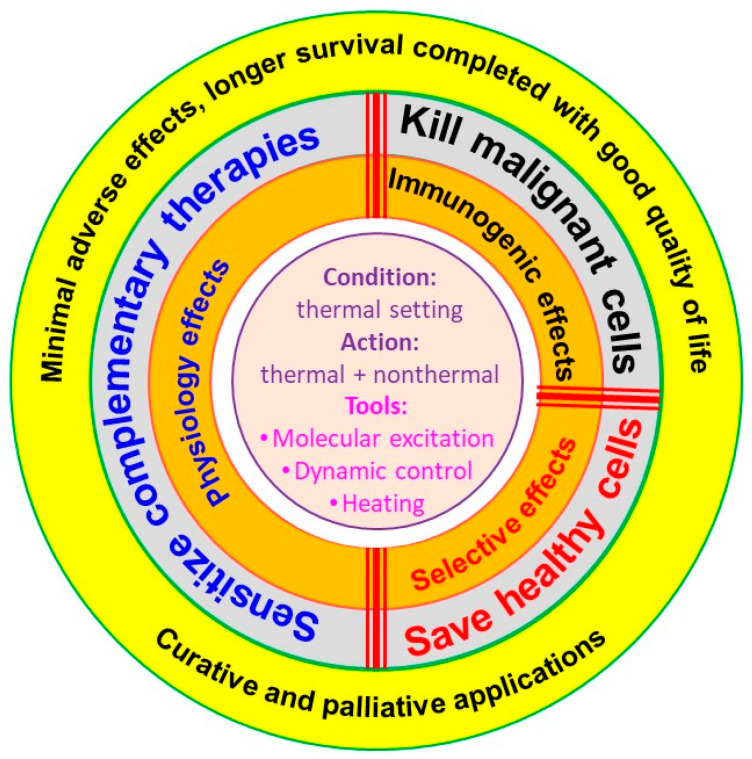
The hallmarks of hyperthermia’s effects in oncology. The main tasks are connected to complementary applications, where the thermal conditions promote the parallel administration of other therapies.

**Figure 3 bioengineering-11-00725-f003:**
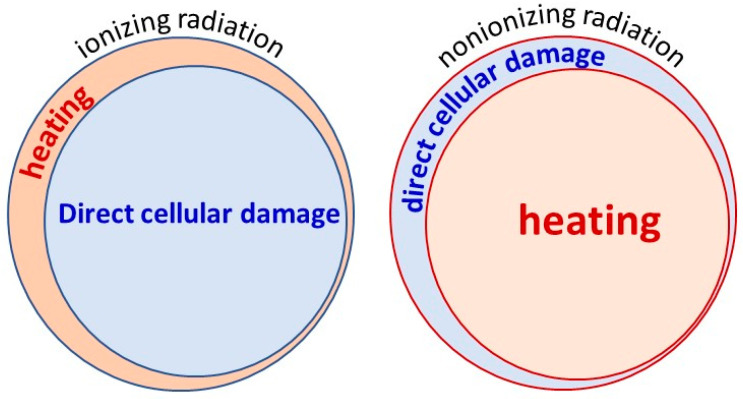
The treatments of electromagnetic radiation sharply differ in ionizing and nonionizing conditions.

**Figure 4 bioengineering-11-00725-f004:**
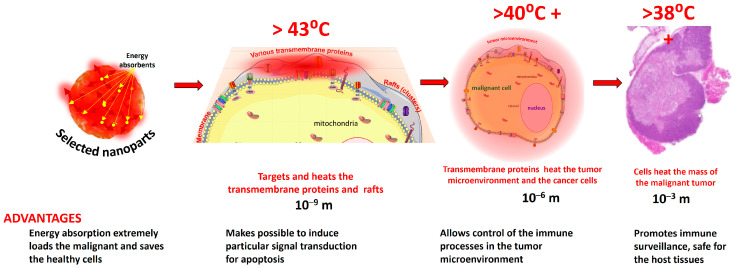
The thermal cascade of the averages for the heterogenic heating of the mEHT. The average sizes magnify about three orders of magnitudes in the different steps.

**Figure 5 bioengineering-11-00725-f005:**
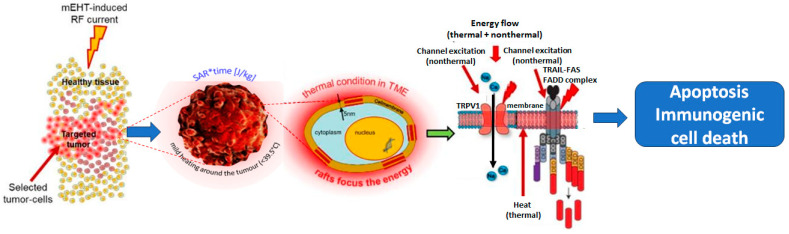
The cell is “gently” destroyed by the support of natural apoptotic signals, allowing the unhurt immunogenic information to be liberated during special apoptosis.

**Figure 6 bioengineering-11-00725-f006:**
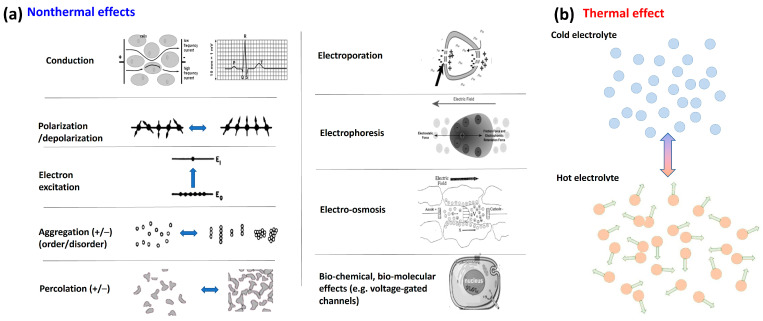
Nonthermal and thermal difference. (**a**) The nonthermal bioelectromagnetic effects vary widely in modifying energy absorption, allowing targeted manipulations of the chemical bonds in the body. (**b**) The thermal effects directly energize the target’s electrolyte components (ions, molecules, cells), significantly increasing their kinetic energy.

**Figure 7 bioengineering-11-00725-f007:**
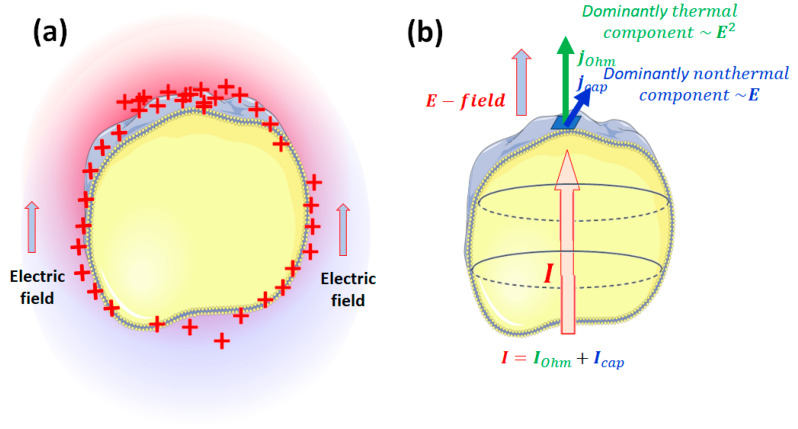
The effects of the external RF field. (**a**) The intense polarization effect of the external field repolarizes the cells, causing hyperpolarization and depolarization states on the membrane. (**b**) The complex current has two components flowing through capacitors (membrane lipid layers).

**Figure 8 bioengineering-11-00725-f008:**
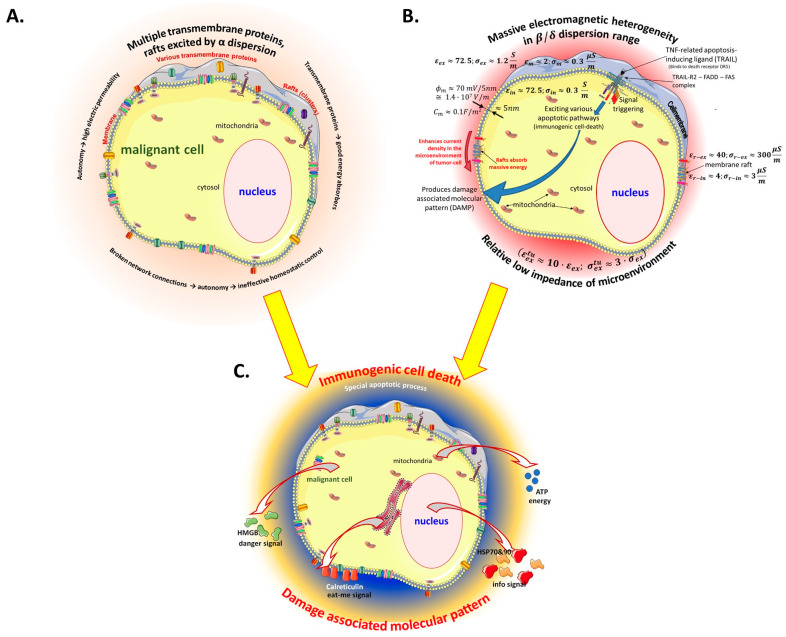
The electromagnetic heterogeneity of the selected tumor cell. (**A**) The transmembrane protein excitations are targeted by α-dispersion. (**B**) The electromagnetic heterogeneity is targeted by particular RF frequency (**C**). The result is the damage associated with molecular patterns induced by immunogenic cell death. Abbreviations/references: $\varepsilon_{ex}$ and $\sigma_{ex}$ are the relative permittivity and conductivity of extracellular electrolytes in the microenvironment of a cell [53]; $\varepsilon_{ex}^{tu}$ and $\sigma_{ex}^{tu}$ are the relative permittivity and conductivity of extracellular electrolytes in the microenvironment of a tumor cell: $\varepsilon_{m}$ and $\sigma_{m}$ are the relative permittivity and conductivity of the cell membrane [54]; $\varepsilon_{in}$ and $\sigma_{in}$ are the relative permittivity and conductivity of intracellular electrolytes of a cell [52]; $\varepsilon_{r-in}$ and $\sigma_{r-in}$ are the relative permittivity and conductivity of the intracellular side of raft proteins [55,56]; $\varepsilon_{r-ex}$ and $\sigma_{r-ex}$ are the relative permittivity and conductivity of the extracellular side of raft proteins [57,58].

**Figure 9 bioengineering-11-00725-f009:**
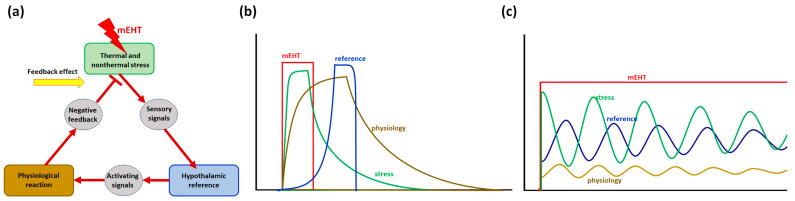
The temperature in the homeostatic range adapts to the new conditions, defining a new equilibrium state. (**a**) Thermal homeostasis has a negative feedback loop as the basis of regulation. (**b**) In a pulse of mEHT, the heat stress rises, and exponential decline returns to the baseline. The thermal reference (hypothalamus) and the physiological counter action (mainly the blood low) have a time lag, acting later. (**c**) In continuing the mEHT impact, the stress, physiology, and reference point fluctuate decreasingly.

**Figure 10 bioengineering-11-00725-f010:**
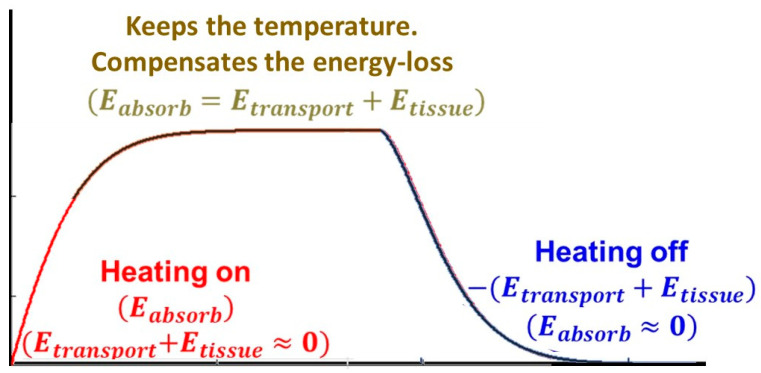
The phases of the temperature development while the absorption is on/off.

**Figure 11 bioengineering-11-00725-f011:**
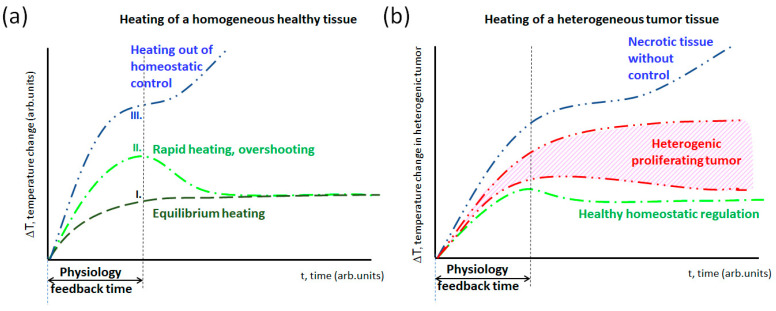
The incident heat could determine the different processes of the control process of thermal homeostasis. (**a**) (I) Is the simplest saturation, a steady state heating of a healthy tissue When the SAR is moderate, the temperature rise is relatively slow (this is the case in most regional treatments). (II) The SAR is high enough for sudden temperature changes, while the physiological thermal feedback only reacts later and regulates the saturation value (this is the case in high-energy local treatments). (III) The SAR is huge; feedback is not able to moderate the temperature, and it toxically burns (this is the case in most ablation treatments). (**b**) The tumor is highly heterogenic, and the temperature develops differently in its parts [71,74]. The high proliferation rate of the tumor enhances the deviation from the healthy equilibrium. A significant volume of the tumor could be necrotic without control.

**Figure 12 bioengineering-11-00725-f012:**
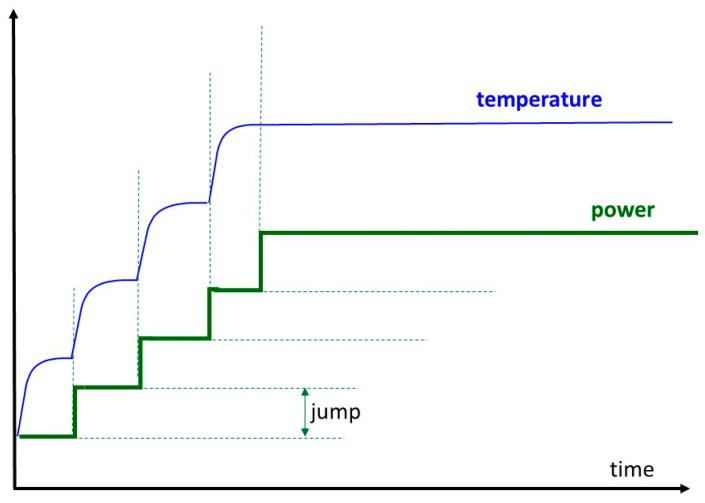
Step-up heating considers physiological adaptation. Step-up heating maintains the steps until homeostatic equilibrium. The provided cumulative energy could vary with the time intervals of the steps.

**Figure 13 bioengineering-11-00725-f013:**
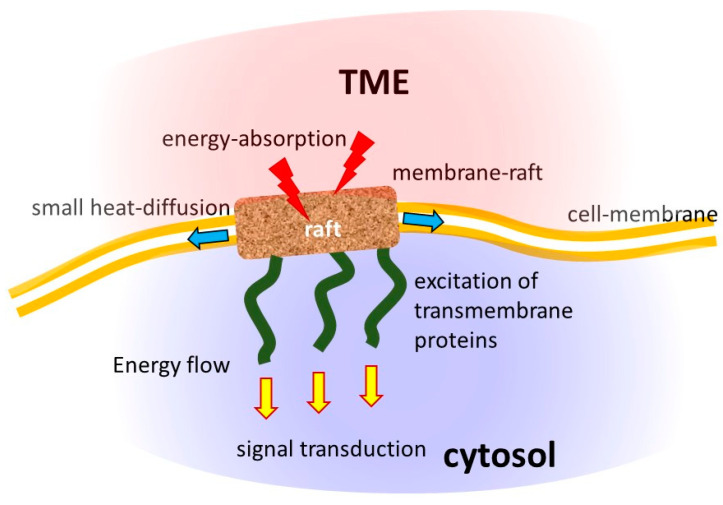
The semi-adiabatic heating of the raft structure. The energy absorption is focused on the rafts. They are embedded in a well-isolating lipid layer, which has bad heat conduction, increasing the time lag of the body’s reaction to the heating process.

**Figure 14 bioengineering-11-00725-f014:**
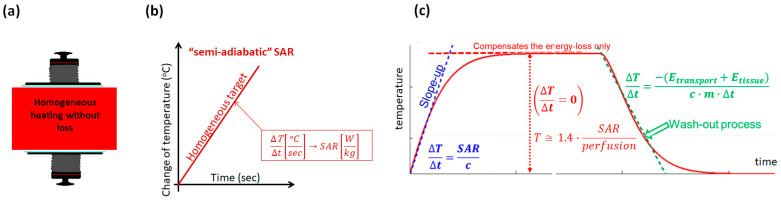
Heating process. (**a**) The target heated homogeneously. (**b**) The starting period heats only the target. It is semi-adiabatic. The physiological feedback has a conditional delay. (**c**) When the thermal regulation makes equilibrium, the temperature does not change, and the perfusion of the electrolyte transfer compensates for the incoming energy. When the power is switched off, the perfusion and tissue heat transfer defines the slope down (wash-out).

**Figure 15 bioengineering-11-00725-f015:**
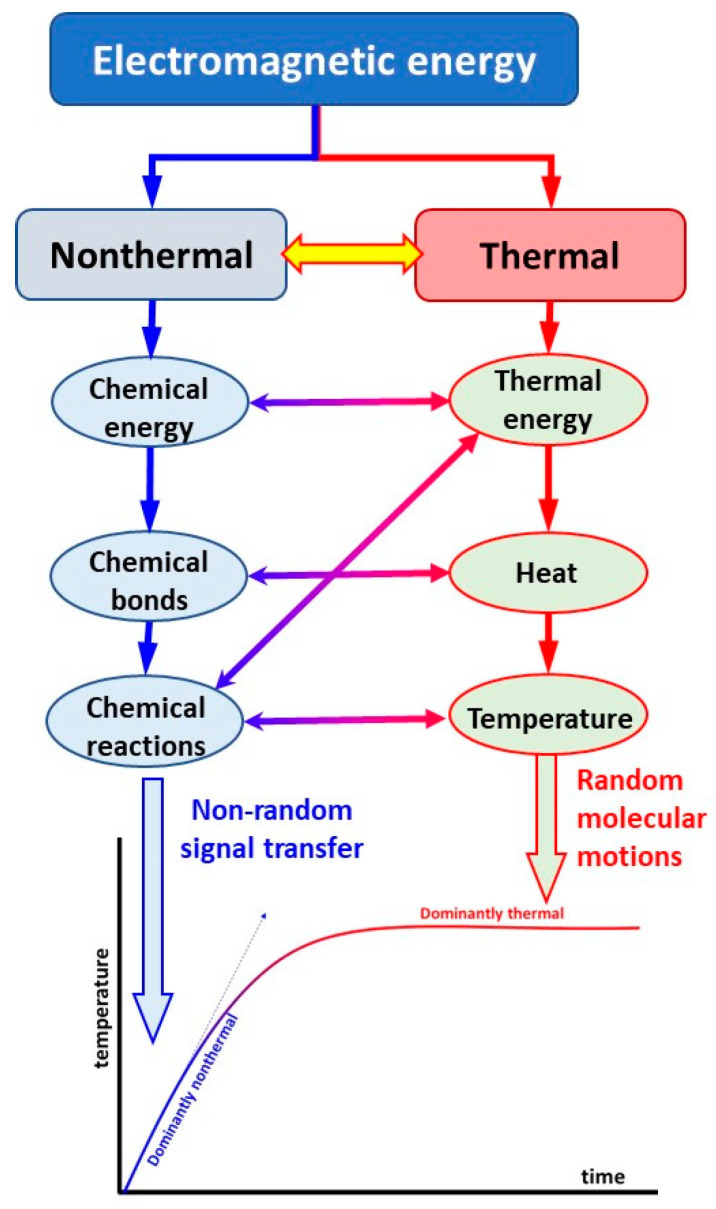
The thermal and nonthermal effects work in synergy. The nonthermal dominates the semi-adiabatic heating period, while the thermal dominates the equilibrium.

**Figure 16 bioengineering-11-00725-f016:**
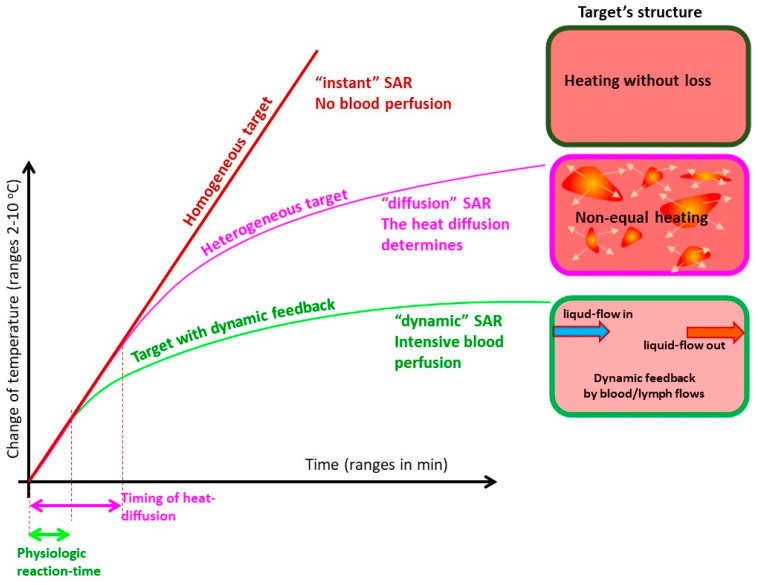
The temperature change strongly depends on the properties of the target. When there is no loss of heat, the target is a homogeneous material and thermally isolated (adiabatic heating); then, the temperature growth is linear, and no equilibrium exists. When only diffusion drives the temperature variation in heterogeneous material, the decline from the adiabatic slope starts, and the system seeks equilibrium at a high temperature. However, when the system has inside transports, the convective energy exchange is added to the conductive one, the radiation cools down the surface, and the dynamic equilibrium appears soon.

**Figure 17 bioengineering-11-00725-f017:**
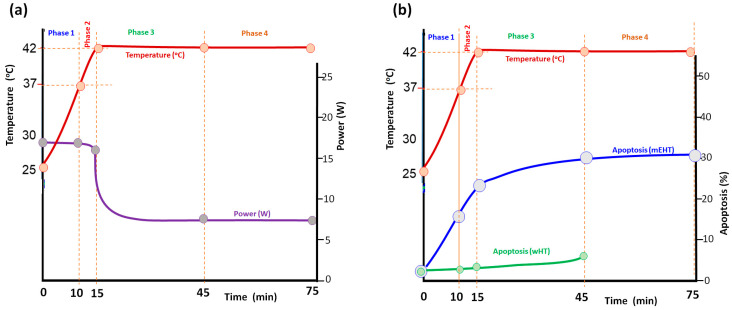
Analyzing the heating process in four phases: heat from 25 °C to 37 °C, from 37 °C to 42 °C, and keep the 42 °C for 30 and 60 min. (**a**) The power and temperature diagram: phase I and II 18 W, phase III and IV 7.5 W, and the absorbed energy in Phase I is 10.8 kJ, in Phase II 5.4 kJ in Phase III 13.5 kJ, and in Phase IV 13.5 kJ [49]. (**b**) The development of apoptosis in mEHT and wHT applications in the A549 cell line.

**Figure 18 bioengineering-11-00725-f018:**
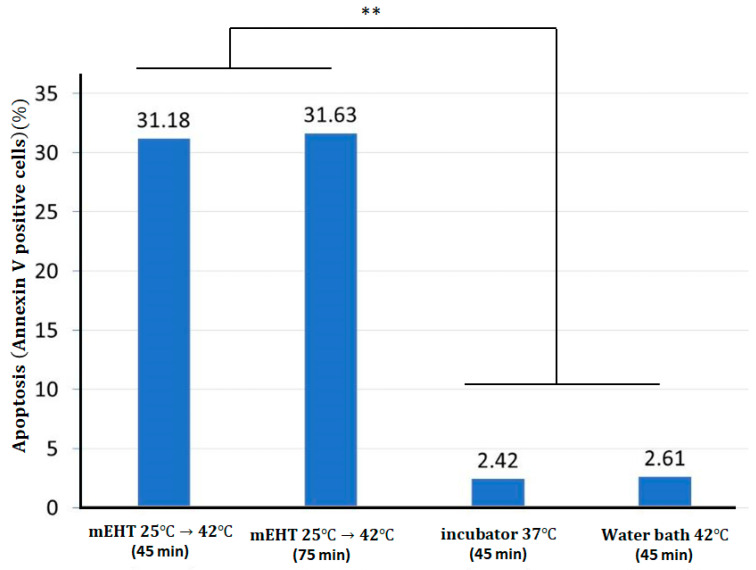
The apoptosis is high in the mEHT, keeping the treated adenocarcinoma human alveolar basal epithelial cell line, A549, at 42 °C, and statistically does not differ from the time of keeping the equilibrium (45 min or 75 min treatments), but the water bath with the same 42 °C has significantly less apoptosis in the same cell line and, at the same time [49]. This difference was observed by others, too [84,86]. It is noteworthy that the process in the incubator at 37 °C has statistically the same result as the water bath at 42 °C. Results are significant (**, *p* < 0.005).

**Figure 19 bioengineering-11-00725-f019:**
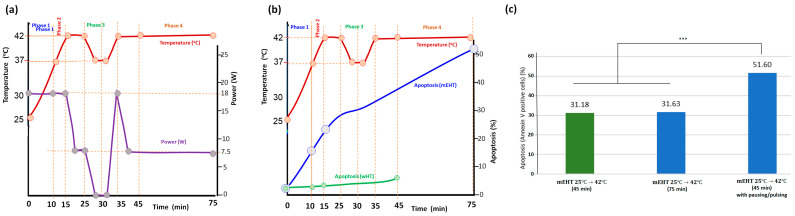
The interruption by pause/pulse power made a significant improvement in apoptosis. (**a**) the power impulse and the consequent temperature change in time. (**b**) The apoptosis development over time with mEHT and wHT is related to temperature. (**c**) Apoptosis in different treatment conditions. The difference between the continuous and pulsed treatment is highly significant (***, *p* < 0.0005).

**Figure 20 bioengineering-11-00725-f020:**
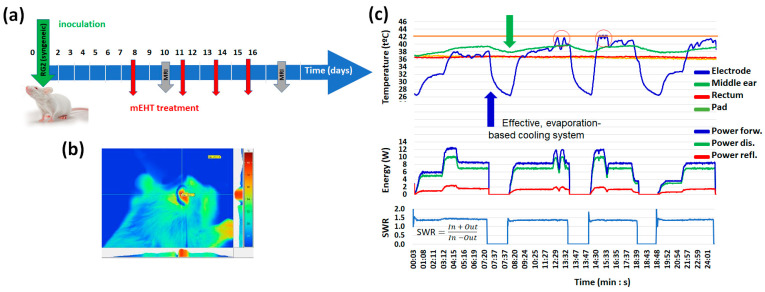
(**a**) The applied protocol. (**b**) Well-localized energy targeting on the head of the rat. (**c**) The parameters during the treatment [93].

**Figure 21 bioengineering-11-00725-f021:**
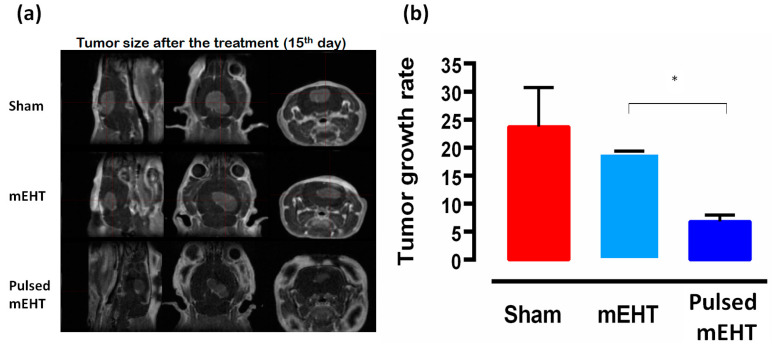
In vivo verification of the advantage of pulsed mEHT. (**a**) MRI imaging of the tumor 15th-day post-treatment with the Mediso nanoScan 1T small animal MRI system and a 3D image acquisition sequence MAGNEVIST^®^, 0.5 mmol/mL, 0.2 mL/kg body. (**b**) Tumor growth rate after the treatment (15th day) [92]. Results are significant (*, *p* < 0.05).

**Figure 22 bioengineering-11-00725-f022:**
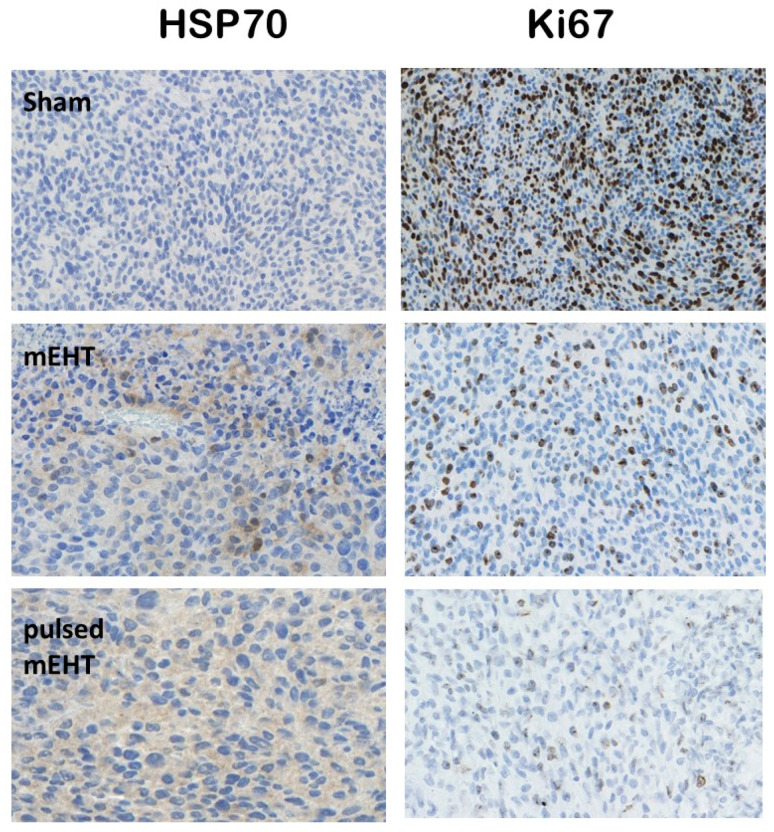
The increase of the extracellular HSP70 and the reduction of the Ki67 proliferation marker significantly increased in the pulsed experiment [92].

**Figure 23 bioengineering-11-00725-f023:**
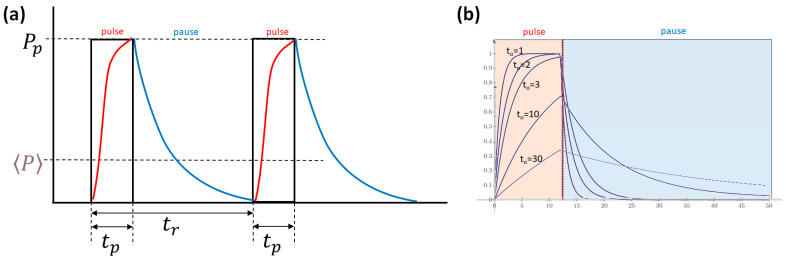
The electric pulsing and the thermal reaction differ. (**a**) The pulse heats the material, which is slower than the electric signal, and the end of the pulse starts a cooling process. (**b**) The thermal parameters of the target ($t_{0}$) define the relaxation time of heating–cooling phases. The change of $t_{0}$ may drastically change the thermal pulse at the same electric pulsing.

**Figure 24 bioengineering-11-00725-f024:**
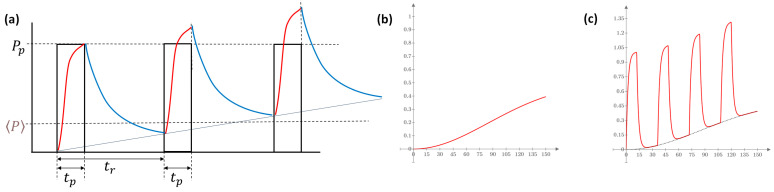
Effect of the duty cycle. (**a**) When the off-period of the pulsing is shorter than necessary for the thermal return to the baseline, the target cumulates the heat, and the overall average temperature grows. (**b**) The average temperature growth can usually be described with the Weibull function. (**c**) The pulsed heating process with Weibull heat-shape.

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
