# Peer review of "Pulsing Addition to Modulated Electro-Hyperthermia"

_bioengineering, 2024, doi:10.3390/bioengineering11070725_

Round 1

Reviewer 1 Report

Comments and Suggestions for Authors

1. “the effects of heating were initially neglected” - line 3. Please cite some effects that were initially neglected.

2. The affirmation on line 36 should be referenced.

3. Remove the dot after the name "Figure 9" - line 251

4. In Figure 8, the labels A, B, and C should be referenced in the legend without the dot. For example, (A.) should be (a). The same is true for other Figures.

6. The equations 1 and 2 are confusing. Please find a better way to display such equations and the meaning of each term. Maybe you can enumerate each one...

7. Lines 281, 282, 283, and 284 have different text styles and sizes. Please check them according to the author's guidelines. Also, all the formatation of the text should be carefully checked.

8. line 297 - what is the meaning of 41C° ? It should be 41 °C?? Please check and put all the units in the S.I.

9. A high percentage (19%) of the text matches with other works of literature. Please check and rewrite it. It shouldn't be more than 3%. 

Comments on the Quality of English Language

The quality of English is ok.

Author Response

Dear Reviewer,

Thank you so much for your suggestions. I followed them all. Here are my responses.

  1. “the effects of heating were initially neglected” - line 3. Please cite some effects that were initially neglected.

Reply: I have done that.

  1. The affirmation on line 36 should be referenced.

Reply: I have referenced it.

  1. Remove the dot after the name "Figure 9" - line 251

Reply: I removed it.

  1. In Figure 8, the labels A, B, and C should be referenced in the legend without the dot. For example, (A.) should be (a). The same is true for other Figures.

Reply: I corrected the labels at all the necessary figures.

  1. The equations 1 and 2 are confusing. Please find a better way to display such equations and the meaning of each term. Maybe you can enumerate each one...

Reply: I rewrote the confusing equations, I hope now they are understandable.

  1. Lines 281, 282, 283, and 284 have different text styles and sizes. Please check them according to the author's guidelines. Also, all the formatation of the text should be carefully checked.

Reply: I am sorry, I did not realize the different styles. Now all is the same.

  1. line 297 - what is the meaning of 41C° ? It should be 41 °C?? Please check and put all the units in the S.I.

Reply: You are right, I accidentally typed it the wrong way. Now it is corrected.

  1. A high percentage (19%) of the text matches with other works of literature. Please check and rewrite it. It shouldn't be more than 3%.

Reply: I checked and there are a couple of references that are my own self references with noted reference numbers.. I would like to keep them please.

I hope you will find the modified manuscript acceptable in its current form now.

Thank you,

Andras Szasz

Reviewer 2 Report

Comments and Suggestions for Authors

The medical application of nonthermal plasma is explored worldwide. In this manuscript, Szasz A summarizes current understanding of medical application of nonthermal plasma hyperthermia. This manuscript is well-organized; however, following point should be corrected.

Minor points

##1: In Figure 20, 21, 22 the reference of original article is not shown. Please show them more clearly.

Author Response

Dear Reviewer,

Thank you so much for your suggestion:

“In Figure 20, 21, 22 the reference of original article is not shown. Please show them more clearly.”

Reply: Thank you for pointing out these missing references. I added them all.

I hope you will find the modified manuscript acceptable in its current form now.

Thank you,

Andras Szasz
